# The Link Between Venous and Arterial Thrombosis: Is There a Role for Endothelial Dysfunction?

**DOI:** 10.3390/cells14020144

**Published:** 2025-01-20

**Authors:** Marco Paolo Donadini, Francesca Calcaterra, Erica Romualdi, Roberta Ciceri, Assunta Cancellara, Corrado Lodigiani, Monica Bacci, Silvia Della Bella, Walter Ageno, Domenico Mavilio

**Affiliations:** 1Department of Medicine and Surgery, Research Center on Thromboembolic Diseases and Antithrombotic Therapies, University of Insubria, 21100 Varese, Italy; walter.ageno@uninsubria.it; 2Centro Trombosi e TAO, Azienda Socio Sanitaria Territoriale dei Sette Laghi, 21100 Varese, Italy; erica.romualdi@asst-settelaghi.it; 3Department of Medical Biotechnologies and Translational Medicine, University of Milan, 20125 Milan, Italy; francesca.calcaterra@humanitasresearch.it (F.C.); roberta.ciceri@humanitasresearch.it (R.C.); cancellara.assunta@hsr.it (A.C.); silvia.dellabella@unimi.it (S.D.B.); domenico.mavilio@humanitas.it (D.M.); 4Unit of Clinical and Experimental Immunology, IRCCS Humanitas Research Hospital, 20089 Rozzano, Italy; 5Center for Thrombosis and Hemorrhagic Diseases, IRCCS Humanitas Research Hospital, 20089 Rozzano, Italy; corrado.lodigiani@humanitas.it (C.L.); monica.bacci@humanitas.it (M.B.); 6Department of Internal Medicine, Ospedale Regionale di Bellinzona e Valli, 6500 Bellinzona, Switzerland

**Keywords:** venous thromboembolism (VTE), arterial thrombosis (AT), endothelial dysfunction (ED)

## Abstract

Venous thromboembolism (VTE) and arterial thrombosis (AT) are distinct yet closely related pathological processes. While traditionally considered separate entities, accumulating evidence suggests that they share common risk factors, such as inflammation and endothelial dysfunction (ED). This review explores the parallels and differences between venous and arterial thrombosis, with particular attention to the role of unprovoked VTE and its potential links to atherosclerosis and systemic inflammation. A key focus is the role of ED, which is emerging as a critical factor in thrombogenesis across both the venous and arterial systems. We examine the current methods for clinically detecting ED, including the use of biomarkers and advanced imaging techniques. Additionally, we discuss novel research avenues, such as the potential of endothelial colony-forming cells and other innovative methodologies, to further unravel the complex mechanisms of thrombosis. Finally, we propose future clinical scenarios where targeting endothelial health could pave the way for more effective prevention and treatment strategies in thrombosis management.

## 1. Introduction

Venous thromboembolism (VTE) and arterial thrombosis (AT) have traditionally been seen as distinct diseases due to differences in their pathophysiology and clinical presentation. While AT is primarily driven by platelet aggregation at sites of atherosclerotic plaque rupture, VTE is associated with stasis, hypercoagulability, and inflammation. However, emerging evidence suggests that these conditions share common risk factors and pathological mechanisms, with endothelial dysfunction (ED) playing a central role. Endothelial cells, which line the blood vessels, regulate vascular tone, maintain hemostasis, and prevent thrombosis. When the endothelium is damaged or dysfunctional, it promotes a prothrombotic environment through impaired anticoagulant mechanisms, increased platelet adhesion, and inflammation. Recent studies have pointed to the overlapping role of systemic inflammation, oxidative stress, and metabolic disorders in both VTE and AT, with ED acting as a potential link between these conditions. This review explores the role of ED in bridging VTE and AT, highlighting its pathophysiological implications and potential clinical impact.

## 2. Venous Thromboembolism and Arterial Thrombosis: Similarities and Differences

Venous and arterial thrombotic disorders have long been considered separate pathophysiological entities. Indeed, AT is considered to originate because of platelet activation [1]. The etiopathogenic mechanism of AT is linked to the presence of atherosclerotic lesions that promote activation of platelets and coagulation by the formation of platelet-rich blood clots, leading to clinically relevant ischemic complications and sometimes death. This process involves two important platelet agonists released at the site of vascular injury: thromboxane A2 (TxA2), generated by cyclooxygenase 1 enzyme/thromboxane synthase activity, and adenosine diphosphate (ADP), which is stored in dense granules. These two agonists act synergistically, amplifying platelet activation and aggregation to facilitate the generation of a stable thrombus with the involvement of the coagulation system [1,2].

VTE, which comprises deep venous thrombosis (DVT) or superficial venous thrombosis (SVT) and pulmonary embolism (PE), is a consequence of the activation of coagulation system along the low-flowing venous system [1,2,3]. VTE is characterized by the formation of thrombi rich in fibrin. This process happens when vascular endothelium is injured, involving the activation of the coagulation cascade with tissue factor as the main mediator that acts synergistically with FVII activated in the site of the lesion, leading to the conversion of prothrombin to thrombin through the prothrombinase complex on activated platelets. Thereafter, platelets adhere to the site of the lesion through their surface glycoprotein GpI-V-IX complex and activate and recruit other platelets through the surface glycoproteins GpIIb/IIIa. This receptor becomes activated upon platelet activation, thus undergoing a conformational change that increases affinity for fibrinogen, which leads to fibrin deposition and the formation of a growing thrombus [1].

However, in addition to this classical pathophysiological characterization of VTE and AT, a considerable amount evidence has suggested that this dichotomy could be an oversimplification, as many studies tend to consider VTE and AT as the aspects of the same disease, thrombosis, sharing mechanisms of action, risk factors, and sometimes the same therapy [2,4] (Table 1).

Considering the mechanisms involved in the onset of an arterial and venous thrombosis, inflammation and endothelial cell damage have been suggested as prior mechanisms involved in the formation of thrombus growth [3,5]. Indeed, in patients with VTE and recurrent thrombosis, an increase in interleukin (IL)-6 and IL-8 and a decrease in anti-inflammatory molecules such as IL-10 were described and directly related with ED [5,6]. Furthermore, the expression of circulating markers as von Willebrand factor, VCAM-1 and soluble p-selectin have been found to be higher in patients with atherosclerotic lesions and venous thrombosis, suggesting inflammation as the cause and not the consequence of thrombosis [3,5]. About this hypothesis, local inflammation of vessel walls has been suggested as the primary event that causes ED and functional deterioration of the circulatory system. Indeed, inflammation triggers the release of cytokines and other mediators that enhance coagulation, inhibit fibrinolysis, and activate platelets [6,7]. Moreover, inflammatory diseases such as rheumatoid arthritis and inflammatory bowel disease are known to increase the risk of both arterial and venous thrombosis, further supporting the hypothesis that inflammation serves as a common thread linking these conditions [8]. In addition, in sickle cell disease, systemic inflammation and thrombotic events have been associated with dysfunction of endothelial nitric oxide (NO) synthase (NOS), causing NO depletion [9]. Finally, COVID-19 was found to be associated with a particularly high rate of venous and arterial thrombosis, explained by endothelial dysfunction/damage and hypercoagulability caused by the strong inflammatory state provoked by SARS-CoV-2, named also “thromboinflammation” [10].

In addition, in the past years, microribonucleic acids (miRNAs), a family of small non-coding RNAs, have been evaluated as a possible diagnostic and prognostic biomarker of vascular diseases [11,12]. Some miRNAs have been well characterized and linked to specific functions; miRNA 126, 155, and 195 amplification or silencing of gene expression are involved in vascular inflammation, cell adhesion, proliferation, and migration processes occurring in the acute phase of VTE and in the progression of arterial disease [11,12]. On the contrary, miRNAs 223 and 96 are related to platelet function; miRNA 223 is involved in P2Y12 regulation and, consequently, in the resistance of anti-platelet P2Y12 drugs, and miRNA 96 is involved in platelet granule secretion, suggesting their hypothetical role in regulating antiplatelet drugs in arterial diseases [12,13].

The clinical potential association between VTE and atherosclerosis was described for the first time in 1995, when Schulman et al. [14], in a 10-year follow-up of a large number of patients with acute VTE, observed a mortality rate associated with acute myocardial infarction (AMI) and stroke significantly higher than that expected in the general population. Subsequently, an association between atherosclerotic disease and VTE of unknown origin was documented. The odds ratios (ORs) for carotid plaques in patients with unprovoked VTE when compared with secondary DVT and controls were 2.3 (95% CI, 1.4–3.7) and 1.8 (95% CI, 1.1–2.9), respectively [4]. Further evidence from clinical studies supported this association between VTE and subsequent cardiovascular events [13,15].

In a systematic review and meta-analysis, Ageno et al. [16] identified factors that increase the risk for VTE and atherosclerosis as obesity, hypertension, diabetes mellitus, and hypertriglyceridemia, but failed to identify smoking as an additional potential risk factor for VTE. Obesity, in particular, is a well-known risk factor for both arterial and venous thrombosis that fosters a pro-inflammatory environment that exacerbates ED and promotes clot formation [17].

Other clinical conditions influence the onset of both arterial and venous thrombosis and constitute additional risk factors, such as hyperhomocysteinemia, antiphospholipid antibodies, malignancies, paroxysmal nocturnal hemoglobinuria, infection states, inflammatory bowel disease, and the use of hormonal therapy. More recently, by an accurate analysis of two population cohort studies, the Emerging Risk Factors Collaboration, with more than 730,000 participants, and the UK Biobank, with more than 420,000 participants, it was determined that only age, male sex, and obesity were associated with an increased risk of VTE complications. Also, it was demonstrated that microalbuminuria, a well-known marker of ED, increases the risk not only of cardiovascular complications but also of VTE disorders; interestingly enough, the risk increases with the levels of albuminuria [18]. Inflammation is likely to play a role as well, and, in this regard, Simes et al., in their recent sub-analysis from almost 8000 patients, reported a strong increase in the risk not only of cardiovascular and all-cause mortality but also of VTE complications in patients with the highest baseline D-dimer values in comparison to those with the lowest ones [19]. In their sub-analysis, Cavallari et al. reported that the rate of VTE complications arising in the follow-up of these patients was strongly related to the number of symptomatic vascular territories involved by atherosclerosis [20]. In addition, the stronger the antiplatelet therapy, the lower the rate of VTE [20].

## 3. The Role of Endothelial Dysfunction in Thrombosis

Endothelium, which lines the interior surfaces of blood vessels, plays a fundamental role in maintaining hemostasis, regulating vascular tone, and modulating the interactions between blood cells and the vessel wall [21]. Additionally, endothelial cells are characterized by high heterogeneity. They differ morphologically, physiologically, and phenotypically along the vascular tree between different organs, as they are responsible for performing several organ-specific functions [22]. When endothelial function is compromised, the delicate balance between pro-thrombotic and anti-thrombotic factors within the vascular system is disrupted, leading to a predisposition for thrombus formation. Indeed, ED is part of Virchow’s triad, which, together with hypercoagulability and hemodynamic alterations, describes what are the main factors responsible for thrombus formation [23].

Under normal conditions, endothelial cells maintain an anticoagulant surface that prevents thrombosis. This is achieved through the expression of various anticoagulant factors such as thrombomodulin (TM), tissue factor pathway inhibitor (TFPI), and heparan sulphate, which together inhibit the coagulation cascade and promote fibrinolysis. Additionally, endothelial cells produce NO and prostacyclin, which have vasodilatory and anti-aggregatory effects, further preventing thrombus formation [24]. However, endothelial cells can also express procoagulant factors, such as tissue factor (TF), von Willebrand factor (vWF), and plasminogen activator inhibitor-1 (PAI-1), and adhesion molecules that mediate endothelial interactions with platelets and immune cells. Their expression is upregulated in activated endothelial cells, thus contributing to clot formation [23,25]. Consequently, the balance between these opposing forces is critical in preventing thrombosis, and any shift towards a procoagulant state that is characteristic of dysfunctional endothelium can lead to a pathologic pro-thrombotic environment by reducing the effectiveness of anticoagulant and fibrinolytic mechanisms [24].

This disruption is particularly relevant in the context of various cardiovascular diseases, where ED is a common underlying pathology, emphasizing the pivotal role of the endothelium in both health and disease. Notably, endothelial cells are surely key players in immune thrombosis, facilitating the interaction between immune cells and the thrombotic process during infections or inflammatory conditions [26]. Nevertheless, factors such as mechanical stress, oxidative damage, and hypercoagulable states can directly impair endothelial function, leading to thrombosis without the involvement of an inflammatory process [27,28].

### 3.1. Mechanisms of ED in Thrombosis

The key features of ED include reduced bioavailability of NO, increased oxidative stress, and the expression of prothrombotic and pro-inflammatory factors. These changes are often associated with cardiovascular risk factors such as hypertension, diabetes, hyperlipidemia, and smoking.

*NO Deficiency and Oxidative stress*. One of the critical mechanisms in ED is the loss of endothelial NO production. NO is a potent vasodilator and inhibitor of platelet aggregation, and its deficiency leads to vasoconstriction, enhanced platelet activation, and increased leukocyte adhesion. These effects contribute directly to a prothrombotic state [29,30]. Oxidative stress further exacerbates this process by reducing NO bioavailability and promoting the production of reactive oxygen species (ROS), which, in turn, amplifies endothelial injury and dysfunction. ROS can oxidize BH_4_ to dihydrobiopterin (BH_2_), limiting NO production [30]. Additionally, the depletion of dihydrofolate reductase impairs the recycling of BH_2_ back to BH_4_, while NADPH oxidase activity exacerbates the reduction in BH_4_ levels, further exacerbating NO uncoupling [30,31,32].

*Upregulation of Adhesion Molecules*. ED is associated with increased expression of adhesion molecules such as P-selectin, E-selectin, VCAM-1, and ICAM-1. These molecules facilitate the adhesion of leukocytes and platelets to the endothelial surface, initiating a local inflammatory response that further promotes thrombosis. This inflammatory milieu induces the expression of TF on endothelial cells and circulating monocytes, triggering the extrinsic pathway of the coagulation cascade and enhancing thrombus formation [23,33].

*Pro-Inflammatory Cytokine Release.* Pro-inflammatory cytokines (i.e., TNF-α, IL-1β, IL-6, IL-8) induce endothelial cell activation. In addition, endothelial cells themselves can release pro-inflammatory cytokines under stress, exacerbating ED and creating a positive feedback loop. This further amplifies the prothrombotic state by enhancing the recruitment of inflammatory cells and the activation of the coagulation cascade [25,34].

*Balance between Tissue Factor (TF) and Thrombomodulin (TM) Expression.* Normally absent in endothelial cells, TF is upregulated in response to inflammatory cytokines such as TNF-α and IL-1β, initiating the extrinsic coagulation pathway, leading to thrombin generation and subsequent clot formation. In contrast, TM, an endothelial membrane protein, acts as a key anticoagulant by binding thrombin and facilitating the activation of protein C, which downregulates clotting activity. Accordingly, under physiological conditions, the equilibrium between TF and TM is crucial to maintaining a balanced hemostatic environment, preventing uncontrolled thrombosis. Disruption of this balance, with TF upregulation and/or decreased TM expression, shifts the system towards a pro-thrombotic state [23,24].

*Dysregulation of Fibrinolysis*. In addition to the inflammatory and procoagulant changes, ED is characterized by an increase in plasminogen activator inhibitor-1 (PAI-1) levels. PAI-1 inhibits tissue plasminogen activator (tPA), a key enzyme in the fibrinolytic pathway. The result is a reduction in fibrinolysis, leading to increased clot stability and persistence. This dysregulation of fibrinolysis is particularly detrimental, as it prevents the natural breakdown of clots and increases the risk of thrombosis becoming chronic and more resistant to resolution [35,36].

#### 3.1.1. ED in Arterial Thrombosis

In the context of AT, ED plays a central role in the initiation and progression of atherosclerosis. The dysfunctional endothelium becomes permeable to lipoproteins, which accumulate in the subendothelial space and undergo oxidation. This triggers a local inflammatory response that attracts monocytes, which differentiate into macrophages and ingest oxidized lipids, forming foam cells. The resulting fatty streaks evolve into atherosclerotic plaques, which can rupture and expose the thrombogenic core of the plaque to the bloodstream, leading to thrombus formation [23,37,38,39].

Moreover, endothelial cells in atherosclerotic regions exhibit reduced anticoagulant activity, further predisposing the site to thrombosis. The decreased expression of TM and tissue factor pathway inhibitor (TFPI) in these areas leads to unchecked thrombin generation, exacerbating the thrombotic risk [39,40].

Additionally, Soeki et al. found that soluble VE-cadherin, a calcium-dependent adhesion molecule critical for maintaining endothelial junction integrity and vascular barrier function, is elevated in patients with coronary artery disease. This elevation is associated with the severity of coronary atherosclerosis, suggesting its involvement in endothelial dysfunction and vascular permeability [41].

#### 3.1.2. ED in Venous Thrombosis

Venous thrombosis is closely linked to ED. In fact, venous thrombosis often results from stasis and hypercoagulability, with ED acting as a crucial mediator. Endothelial cells in veins typically prevent thrombosis by producing anticoagulant factors, such as NO and prostacyclin, and expressing surface molecules like TM and heparan sulfate proteoglycans that inhibit clot formation. When ED occurs, these protective mechanisms are impaired, tipping the balance toward a prothrombotic state. This dysfunction can also be further induced by factors such as inflammation, hypoxia, or mechanical stress from blood stasis [25,42,43].

A critical event in venous thrombosis is the expression of TF by dysfunctional endothelial cells, initiating the coagulation cascade. Additionally, upregulation of adhesion molecules, such as P-selectin and E-selectin, promotes leukocyte and platelet recruitment to the endothelial surface, further propagating thrombus formation [23,33].

Hypoxia, often a consequence of venous stasis, exacerbates ED by upregulating prothrombotic factors like PAI-1 and vWF while downregulating fibrinolytic activity. This prothrombotic environment not only promotes thrombus formation but also impairs thrombus resolution, leading to persistent and potentially life-threatening conditions such as pulmonary embolism or chronic thromboembolic pulmonary hypertension [35,36,44,45].

## 4. Classical and Novel Biomarkers of Endothelial Dysfunction

Despite significant advances in understanding cardiovascular risk factors, cardiovascular mortality rates remain persistently high. This underscores the importance of accurate and early assessment of ED as a crucial prognostic tool for predicting cardiovascular events. ED is a critical early indicator of vascular health, and its evaluation can provide valuable insights into the risk of thrombotic events, which are major contributors to cardiovascular morbidity and mortality.

Classical markers of ED used in vascular research include circulating levels of pro-inflammatory molecules and soluble adhesion molecules, such as E-selectin, ICAM-1, VCAM-1, and VE-cadherin, which are used to assess the integrity of endothelial function. Additionally, molecules involved in the coagulation pathway, such as vWF and soluble TM, are commonly evaluated [46].

Some of these conventional biomarkers have been extensively studied in large-scale clinical trials and epidemiological studies, solidifying their role in cardiovascular risk assessment. For instance, elevated levels of C-reactive protein, IL-6, ICAM-1, VCAM-1, and E-selectin have been consistently linked to various atherosclerotic manifestations, including peripheral vascular disease, ischemic heart disease, and acute myocardial infarction. Notably, adhesion molecules like ICAM-1 and VCAM-1 have shown promise in predicting the onset of coronary heart disease [46,47,48,49].

However, there has been insufficient evidence so far on the diagnostic accuracy and predictive power of these markers to allow their introduction in common clinical practice. Moreover, the complexity of endothelial function suggests that relying solely on these conventional biomarkers may not provide a comprehensive picture. Many of these markers are not exclusively generated by endothelial cells, which limits their specificity for assessing endothelial damage or dysfunction. Thus, there is a growing need to identify additional biomarkers that, alone or in combination with conventional markers, can more accurately reflect the state of the endothelium and predict thrombotic risk.

To address the limitations of conventional biomarkers, new soluble and cellular biomarkers of ED are gaining attention for their potential role in evaluating ED and thrombotic events.

### 4.1. Soluble Biomarkers

*Endocan*, also known as endothelial cell-specific molecule-1 (ESM-1), is a soluble proteoglycan primarily expressed by endothelial cells. It plays a significant role in the modulation of endothelial function and inflammatory processes [50]. In cardiovascular diseases, elevated levels of endocan are associated with ED, inflammation, and atherosclerosis [50]. Endocan is involved in various pathological processes, including the regulation of cell adhesion, migration, and proliferation. It can bind to integrins and other adhesion molecules, influencing leukocyte recruitment to the endothelium and contributing to the inflammatory response in vascular diseases [51,52,53]. Moreover, endocan expression is upregulated by pro-inflammatory cytokines such as TNF-α and IL-1β, further linking it to the inflammatory pathways involved in cardiovascular pathology [50,54]. In clinical settings, elevated serum levels of endocan have been proposed as a biomarker for ED and a predictor of adverse cardiovascular events, including myocardial infarction and stroke [50,51,52,53]. Studies also indicate that endocan levels correlate with thromboembolic diseases. For instance, patients with massive or submassive PE show significantly higher endocan levels than those with nonmassive PE or healthy controls, these findings suggesting endocan’s potential as a biomarker for assessing PE severity and monitoring thrombolytic therapy [55]. Furthermore, while plasma endocan levels alone may not reliably distinguish DVT from other inflammatory conditions, combining it with other biomarkers could improve diagnostic accuracy [56]. Its role in vascular diseases underscores the potential of targeting endocan in therapeutic strategies aimed at reducing inflammation and improving endothelial function.

*Endoglin* (*CD105*) is a crucial transmembrane glycoprotein predominantly found on proliferating endothelial cells. It acts as a co-receptor for transforming growth factor-β (TGF-β), a key regulator of endothelial cell proliferation, angiogenesis, and vascular integrity [57]. Endoglin exists in two main isoforms: long-form (L-endoglin) and short-form (S-endoglin). While L-endoglin supports endothelial cell repair and enhances TGF-β signaling, S-endoglin acts as a negative regulator, inhibiting this pathway, thus conferring to L-endoglin a proangiogenic role and to S-endoglin an antiangiogenic role [58,59]. Moreover, L-endoglin enhances NO-dependent vasodilation by increasing Smad2 protein levels and NOS expression, while S-endoglin impairs this vasodilation, though the molecular mechanisms remain unclear [58,60]. Elevated levels of S-endoglin have been linked to various cardiovascular conditions such as hypertension, hypercholesterolemia, and diabetes, suggesting its role as a marker of early vascular dysfunction [61,62]. Despite the fact that high levels of soluble endoglin were described in several cardiovascular diseases, and that increasing evidences showed that endoglin and its circulating form are involved in hemostasis and processes of thrombo-inflammation, endoglin’s role in patients who experienced thrombosis is still unclear. In this context, Rossi and colleagues showed, in a mouse model, that membrane-bound endoglin has a role in hemostasis contributing to platelet recruitment through interaction with αIIbβ3. On the contrary, circulating endoglin seems to act as a competitor of membrane-bound endoglin in the binding to platelet αIIbβ3, thus interfering with thrombus formation and stabilization [63].

*Asymmetric dimethylarginine* (*ADMA*) is an endogenous inhibitor of NOS, the enzyme responsible for the production of NO from L-arginine [64]. Elevated levels of ADMA are associated with reduced NO production, leading to ED that results in increased vascular resistance, higher blood pressure, and a pro-thrombotic state. The constituted ED is a precursor to a variety of cardiovascular diseases, including hypertension, atherosclerosis, and heart failure [65,66,67]. ADMA is metabolized by dimethylarginine dimethylaminohydrolase (DDAH), and its activity is crucial for maintaining ADMA levels within a physiological range. However, in cardiovascular diseases, the activity of DDAH is often reduced, leading to elevated ADMA levels [29,65]. Importantly, recent findings indicate that elevated plasma ADMA levels are predictive of VTE, suggesting that ADMA contributes not only to arterial but also venous thrombotic processes [68]. ADMA is considered both a biomarker and a therapeutic target in the prevention and treatment of cardiovascular diseases [69].

### 4.2. Cellular Biomarkers

*Endothelial microparticles* (*EMPs*) are small vesicles that are derived from endothelial cells and play a crucial role in cardiovascular diseases by contributing to ED, inflammation, and thrombosis [70]. EMPs are characterized by surface proteins from endothelial cells, such as VE-cadherin, PECAM-1, ICAM-1, VCAM-1, E-selectin, and others, like endoglin and TF [71]. EMPs are primarily released in response to cell activation or apoptosis, often triggered by various stimuli such as oxidative stress, inflammatory cytokines, or shear stress [70,71]. They are associated with a pro-inflammatory and pro-thrombotic state, making them potential biomarkers for endothelial injury and cardiovascular risk [71,72]. EMPs can also impair endothelial function by reducing NO bioavailability, which is essential for vascular relaxation and protection against atherosclerosis [73]. High levels of endoglin-positive EMPs have been linked to ED, especially in conditions like preeclampsia [74].

In addition to EMPs, other strategies for ED assessment that apply cellular biomarkers consist of the quantification of *circulating endothelial cells* (CECs) and the quantification of *circulating endothelial progenitor cells* (EPCs) by flow cytometry [75]. CECs derive from the detachment of mature ECs from the vessel intima, and their amount in circulation is increased in the case of EC activation or injury, as in patients with thrombotic diseases, vascular inflammation, atherosclerosis, and COVID-19 [76]. Concerning EPCs, these endothelial progenitors are mobilized in response to stimuli such as ischemia, including myocardial infarction, ischemic stroke, vascular trauma, sickle cell anemia, vasculitis, and pulmonary hypertension. EPCs are typically identified on the basis of absent/weak CD45 expression (unlike hematopoietic cells) combined with the expression of immaturity (CD34) and endothelial markers (CD146, VEGFR-2/KDR) [77]. In several studies, it has been shown that EPC levels seem to predict the occurrence of cardiovascular events and death from cardiovascular causes and may help to identify patients at increased cardiovascular risk [78]. In the context of thrombosis, the ExACT study showed that patients with a higher level of circulating EPCs, measured at least three months after a treated episode of unprovoked VTE, were significantly less likely to develop VTE recurrence [77].

In order to study more in depth, the mechanism(s) underlying ED, there is a growing scientific interest in using patient-derived endothelial colony-forming cells (ECFCs) as a non-invasive approach to study endothelial alterations. ECFCs are endothelial progenitors that can be isolated and cultured in vitro from peripheral blood; they are endowed with the capacity to proliferate and form blood vessels, playing a key role in vascular repair and regeneration [79]. In patients with thrombotic disorders, quantitative and functional alterations of ECFCs have been observed, making them a promising biomarker for ED. Monitoring ECFC levels and functionality could provide valuable information on endothelial health and the likelihood of thrombotic events [79,80,81]. In this scenario, we recently identified the pathologic upregulation of the TNFSF15–TNFRSF25 axis in patients with unprovoked VTE, which impairs ECFC function and may contribute to the development of VTE [82]. Moreover, in a further study that is in progress, we observed the presence of ED features in ECFCs obtained from a second cohort of unprovoked VTE patients but not in ECFCs obtained from patients with provoked VTE [83]. Notably, in these studies, ECFCs were isolated from blood samples collected months after the acute event, thus allowing the study of primary ED by avoiding the confounding effects due to the thrombotic event. Additionally, their regenerative properties position ECFCs as both potential biomarkers and therapeutic targets for restoring endothelial function in different pathological contexts [81,84,85,86,87].

### 4.3. Multi-Biomarker Approach and Future Directions

Considering endothelial heterogeneity [22], and given the multifaceted nature of ED and thrombotic events, a multi-biomarker approach that combines endocan, endoglin, ADMA, EMPs, CECs, and conventional markers could offer a more comprehensive assessment. This strategy could improve patient stratification, allowing for more personalized and effective management of thrombotic risk (Figure 1). In addition, the timing of ED evaluation is critical. Assessments conducted during the acute phase help identify ED resulting from thrombosis, whereas evaluations performed after at least three months from the acute event can provide insights into primary ED, which may be a contributing cause rather than a consequence of thrombosis. Identifying primary ED and determining its persistence after thrombus resolution is essential for understanding its role in the recurrence of thrombosis.

Furthermore, future research should focus on standardizing the measurement of these novel biomarkers and validating their clinical utility across diverse patient populations. Understanding the interplay between these biomarkers and the mechanisms underlying thrombotic events will also be essential for identifying new therapeutic targets and improving cardiovascular outcomes.

## 5. Clinical Detection of Endothelial Dysfunction by Imaging Testing

Imaging modalities that assess endothelial function, particularly noninvasive techniques, have become essential for early detection and monitoring of ED (Figure 1). One of the most employed methods is flow-mediated dilation (FMD), which uses high-resolution ultrasound to measure the dilation of the brachial artery in response to increased blood flow. This method assesses the release of NO and the vasodilation it mediates, providing a proxy for endothelial health [43,88].

Reduced FMD values are associated with an increased risk of cardiovascular events and may reflect systemic endothelial impairment, correlating with both arterial and venous thrombotic risks [89,90].

Radiofrequency-based vascular ultrasound and wall-tracking systems (WTSs) offer additional insights, particularly for arterial stiffness measurements, which are influenced by endothelial health. These techniques capture arterial wall movements with high spatial and temporal resolution, allowing clinicians to assess carotid intima-media thickness (IMT) and arterial stiffness through parameters like pulse wave velocity (PWV) and beta stiffness index. These measurements have been validated as predictors of cardiovascular risk, especially in patients with hypertension and atherosclerosis, and may be useful in evaluating patients at risk of VTE [91,92].

Emerging tools such as advanced carotid ultrasound imaging and wall-tracking systems provide refined measures of endothelial function and arterial mechanics. For example, carotid IMT and local carotid stiffness indices can reflect age-related or disease-specific changes in vascular health. These parameters, when combined with FMD, give a more comprehensive view of endothelial function, potentially aiding in the identification of patients at risk for both arterial and venous thrombosis [89,93].

## 6. Future Clinical Scenarios

From a clinical standpoint, the recognition of the association between atherosclerosis, inflammation, and VTE carries significant implications.

A future diagnostic approach that combines biomarkers, endothelial function imaging, and cellular indicators like EPCs, ECFCs, and EMPs could offer a comprehensive assessment of ED. These strategies may not only improve early detection and risk stratification in VTE and atherosclerosis but also pave the way for targeted therapeutic interventions aimed at preserving endothelial function and reducing thrombotic complications.

In particular, targeting modifiable inflammatory and metabolic risk factors, such as obesity and smoking, could play a dual role in preventing the recurrence of VTE and the progression of atherosclerosis.

Pharmacological interventions may also help reduce the inflammatory burden associated with both conditions. Statins, for example, are widely used in the treatment of atherosclerosis for their lipid-lowering and anti-inflammatory effects, and emerging evidence suggests they may also reduce the incidence of first episode and recurrent VTE [94,95,96]. The anti-inflammatory properties of statins might directly influence the shared mechanisms underlying both VTE and atherosclerosis. Indeed, beyond their cholesterol-lowering action, statins improve endothelial function by enhancing NO bioavailability and reducing oxidative stress, which can stabilize the endothelium and reduce thrombosis risk [97].

Moreover, therapies that specifically target inflammatory pathways could be impactful. Agents like interleukin-6 (IL-6) inhibitors, which have demonstrated efficacy in reducing cardiovascular events [98] may also help in lowering VTE risk by reducing endothelial activation. Selective anti-inflammatory agents could reduce systemic inflammation, a key factor in ED, and offer targeted protection against thrombosis.

In the future, cellular therapies based on ECFCs may offer exciting possibilities for repairing damaged endothelium. These cells play a role in vascular repair and regeneration, and their administration could help restore endothelial function in patients with significant endothelial injury. Preliminary studies indicate that increasing EPC or ECFC levels could enhance vascular health, potentially lowering the recurrence risk of VTE and atherosclerosis progression. Nevertheless, several challenges must be overcome before moving ECFCs into human subjects as cell therapy. Among these, we can find the definition of distinguishing markers for the ECFCs with high proliferative potential and strategies to overcome the limited availability of autologous ECFCs. In this context, generation of patient-specific human induced pluripotent stem cell (hiPSC)-derived ECFCs and/or haplotype-based banking of hiPSCs to generate ECFCs compatible with recipients are strategies under evaluation [99].

Finally, new therapeutic targets such as endoglin and endocan—biomarkers of endothelial activation and inflammation—as well as molecules involved in pathogenetic mechanisms of ED identified through ECFC characterization, offer potential for anti-thrombotic targeted therapies. Blocking these molecules may reduce endothelial activation and subsequent thrombus formation. Drugs targeting these molecules are in development and could provide highly specific treatments for preventing ED-related thrombosis [100,101].

Moreover, there are several aspects of endothelial homeostasis and dysfunction that need to be elucidated. Indeed, endothelial cell characteristics vary across different sites of the vascular system and, in the same organ, even across vessels (post-capillaries, venules, arterioles, arteries, small vessels, and large vessels). From a clinical standpoint, thrombotic events show peculiar clinical features when occurring at unusual sites (e.g., splanchnic vein thrombosis or cerebral vein thrombosis as compared to typical DVT of the lower limbs) [102]. Therefore, the role of ED and the possibility of using it as a diagnostic tool and/or therapeutic maker need to be further investigated and confirmed across different vascular beds.

## 7. Conclusions

The relationship between VTE, atherosclerosis, and ED underscores the complex interplay between venous and arterial systems. Inflammation and ED emerge as central, shared mechanisms that drive both arterial and venous thrombogenesis, suggesting that these vascular disorders may be interlinked rather than isolated. Key mechanisms of ED include reduced NO bioavailability, oxidative stress, and increased expression of pro-thrombotic and inflammatory mediators. These alterations disrupt the endothelium’s balance between coagulation and anticoagulation, contributing to a heightened thrombotic state. Biomarkers of ED, such as circulating levels of von Willebrand factor, ICAM-1, and P-selectin, and imaging evidence of ED, as detected by FMD and arterial stiffness evaluation, can provide valuable insights into vascular health and thrombotic risk. Emerging biomarkers, like EMPs and ECFCs, offer promising tools for detecting endothelial injury and dysfunction.

This comprehensive understanding of ED’s role opens avenues for targeted interventions to improve prevention and management strategies in patients with unprovoked VTE and atherosclerosis. By identifying and mitigating inflammation and other modifiable risk factors, there is potential to reduce the recurrence of thrombotic events and improve long-term outcomes. This integrative approach to endothelial health highlights ED as both a diagnostic target and a therapeutic focus, with significant implications for personalized vascular care.

## Figures and Tables

**Figure 1 cells-14-00144-f001:**
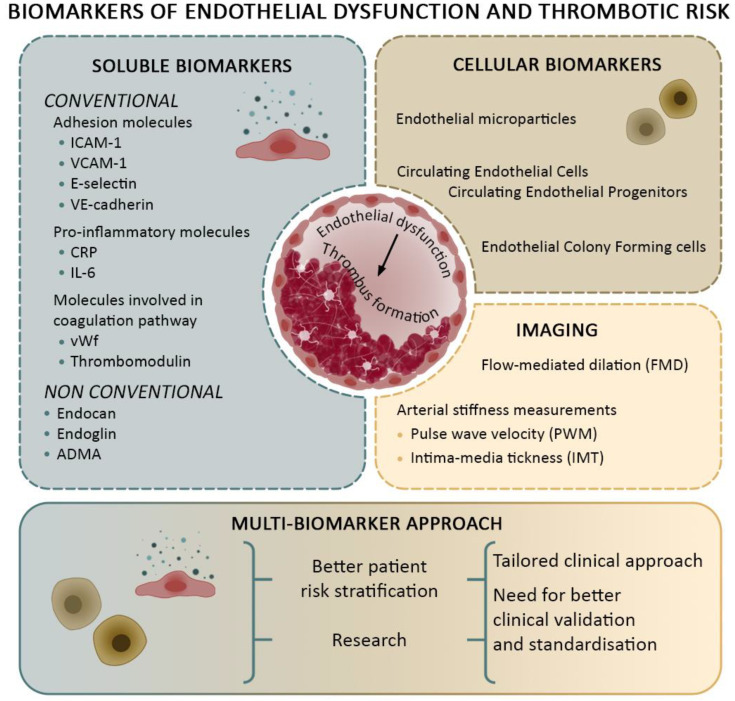
Multi-biomarker approach to assess endothelial dysfunction. Scheme summarizing the biomarkers that can be assessed to evaluate endothelial dysfunction and thrombotic risk by applying a multi-biomarker approach. Bio-markers are subdivided into soluble biomarkers, cellular biomarkers, and imaging approaches.

**Table 1 cells-14-00144-t001:** Similarities and differences between arterial and venous thrombosis.

	Arterial Thrombosis	Venous Thrombosis	Similarities
Epidemiology	Commonest cardiovascular disease (approximately 3–6 cases per 1000 persons per year)	3rd most common cardiovascular disease (approximately 1–2 cases per 1000 persons per year)	Very common causes of vascular disease
Primary Pathogenesis	Primary platelet activation	Primary coagulation cascade activation	Thrombus formation
Composition of clots	Platelet-rich with limited fibrin (white thrombi).	Rich in fibrin and red blood cells (red thrombi).	Interplay between platelet activation and coagulation cascade
Pathophysiology	Atherosclerotic plaque complications causing platelet adhesion	Coagulation around vein valves activated by venous stasis, hypercoagulability, and endothelial damage (Virchow’s triad)	Endothelial cells damaged/activated by inflammation, oxidative stress, and metabolic disorders
	Tissue ischemia and infarction	Pulmonary embolization	Abnormal clot that obstructs blood flow and can result in ischemic damage to tissues downstream
Major risk factors	Smoking, hypertension, diabetes, dyslipidemia	Major surgery, major trauma, cancer, puerperium, acute illness with immobilization, thrombophilia	Several risk factors overlap (increasing age, cancer, obesity, dyslipidemia, inflammatory disorders)
Most common locations	Brain, heart, lower limbs	Lower limbs, lung, upper limbs	Virtually any vascular bed can be involved
Clinical manifestations	Signs and symptoms of tissue ischemia and infarction (depending on location: chest pain, stroke symptoms, lower limb pain and paleness)	Signs and symptoms of venous obstruction and/or embolization (depending on location: leg swelling, pain, redness, dyspnea, chest pain)	Emergency conditions requiring rapid diagnosis and treatment
Mortality	Leading cause of death globally, especially from myocardial infarction and stroke (case fatality rate around 10–20%)	Case fatality rate associated with pulmonary embolism around 10%	Commonest cause of morbidity and mortality worldwide

## Data Availability

Not applicable.

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
