# Peer review of "The Link Between Venous and Arterial Thrombosis: Is There a Role for Endothelial Dysfunction?"

_cells, 2025, doi:10.3390/cells14020144_

Round 1
Reviewer 1 Report
Comments and Suggestions for Authors
Comment for the authors:
1. Please introduce the acronym "AT" for arterial thrombosis in the first paragraph of the manuscript to ensure clarity and consistency.
2. Include a table summarizing the similarities and differences between AT and venous VTE. This will help readers quickly grasp these two conditions' key distinctions and overlaps.
3. The acronym "ED" is not defined in the manuscript. If it refers to endothelial dysfunction, please clarify this in the text where it is first mentioned.
4. The manuscript does not address the disruption of endothelial cell permeability as a critical feature of atherosclerosis. Additionally, soluble VE-cadherin has been suggested as a potential marker of coronary artery disease (reference: Circ J. 2004 Jan;68(1):1-5. doi: 10.1253). Consider incorporating this aspect into the discussion to provide a more comprehensive overview of endothelial dysfunction in atherosclerosis.
Author Response
- Please introduce the acronym "AT" for arterial thrombosis in the first paragraph of the manuscript to ensure clarity and consistency.
In accordance with Reviewer’s suggestion, the acronym “AT” was introduced in the abstract and in the first paragraph of the manuscript (please see on page 1, line 38).
- Include a table summarizing the similarities and differences between AT and venous VTE. This will help readers quickly grasp these two conditions' key distinctions and overlaps.
We thank the Reviewer for this suggestion, we have added a Table summarizing the similarities and differences between the arterial and venous thrombosis.
- The acronym "ED" is not defined in the manuscript. If it refers to endothelial dysfunction, please clarify this in the text where it is first mentioned.
We thank the Reviewer for this indication and we apologize for not defining ED where it was first mentioned. Therefore, the acronym “ED”, that refers to endothelial dysfunction, was introduced in the abstract and in the first paragraph of the revised manuscript (please see on page 1, line 44)
.4. The manuscript does not address the disruption of endothelial cell permeability as a critical feature of atherosclerosis. Additionally, soluble VE-cadherin has been suggested as a potential marker of coronary artery disease (reference: Circ J. 2004 Jan;68(1):1-5. doi: 10.1253). Consider incorporating this aspect into the discussion to provide a more comprehensive overview of endothelial dysfunction in atherosclerosis.
We agree with the Reviewer that endothelial cell permeability disruption plays a key role in atherosclerosis. Accordingly, we discussed this aspect in section 3.1.1 also citing the suggested work (please see the revised manuscript on page 5, lines 249-253). We thank the Reviewer since his/her comment helped us to improve this section.
Reviewer 2 Report
Comments and Suggestions for Authors
Arterial and venous thrombosis have traditionally been considered as distinct entities. This manuscript highlights the role of endothelial dysfunction ED as an emerging central risk factor common to both types of thrombosis.
The manuscript is clear and extensively documented by an appropriate literature.
After a thorough description of classical biomarkers of ED, its originality and main contribution lies in a description of new biomarkers as main exploratory targets common to both types of thrombosis and suggests new therapeutic perspectives.
Minor remarks
editorials: lots of inappropriate spaces between words and sometimes cutting off certain words throughout the manuscript => to editors
Lane 123, cohort studies
Other comments
First part of the manuscript: ED dysfunction has been described in COVID 19 as common risk factor of both arterial and venous thrombosis
In the same way, in sickle cell disease, ED driven by inflammation and NO depletion seems to pay a central role in arterial and presumably predisposition to venous thrombosis
Second part of the manuscript: Endocan, endoglin and ADMA/DDAH seems to be only documented, at least till now in cardiovascular diseases. Any perpectives in VTE? The composition of the vascular tree being different in the arterial and venous network, could the properties of the endothelial cells vary from one to the other
Lane 324 “Postnatal EPCS” could you clarify this term postnatal?
Lane 412 “administration of EPC and ECFCs could help restore endothelial cell function”
Do you mean transplantation of exogenous endothelial cells?
Author Response
Arterial and venous thrombosis have traditionally been considered as distinct entities. This manuscript highlights the role of endothelial dysfunction ED as an emerging central risk factor common to both types of thrombosis.
The manuscript is clear and extensively documented by an appropriate literature.
After a thorough description of classical biomarkers of ED, its originality and main contribution lies in a description of new biomarkers as main exploratory targets common to both types of thrombosis and suggests new therapeutic perspectives.
We thank the Reviewer for the endorsement of our work.
Minor remarks
editorials: lots of inappropriate spaces between words and sometimes cutting off certain words throughout the manuscript => to editors
We revised some typos and remove inappropriate spaces throughout the manuscript.
Lane 123, cohort studies
We revised it in the manuscript.
Other comments
First part of the manuscript: ED dysfunction has been described in COVID 19 as common risk factor of both arterial and venous thrombosis.
In the same way, in sickle cell disease, ED driven by inflammation and NO depletion seems to pay a central role in arterial and presumably predisposition to venous thrombosis
We thank the Reviewer for these comments that enabled us to improve the manuscript. We have added some sentences on ED dysfunction in COVID 19 and in sickle cell disease. Please find them in the revised manuscript (page 2, lines 101-107).
Second part of the manuscript: Endocan, endoglin and ADMA/DDAH seems to be only documented, at least till now in cardiovascular diseases. Any perpectives in VTE?
We thank the Reviewer for raising this interesting point that deserved to be discussed more in depth. In the revised manuscript, the section “4.1 Soluble Biomarkers” was implemented adding information about the above-mentioned markers also in the context of venous thrombosis. (Please see the revised manuscript on page 6 (lines 318-325), page 7 (lines 340-348 and lines 358-360).
The composition of the vascular tree being different in the arterial and venous network, could the properties of the endothelial cells vary from one to the other
We thank the Reviewer for this insightful comment. Indeed, endothelial cells characteristics vary across different sites of the vascular system and, in the same organ, even across vessels (post-capillaries, venules, arterioles, arteries, small vessels, and large vessels). Moreover, even thrombotic events show peculiar clinical features when occurring at unusual sites (eg. splanchnic vein thrombosis or cerebral vein thrombosis as compared to typical DVT of the lower limbs). Even if we focused on the typical manifestations of thrombosis, we have now mentioned these points in the revised manuscript (please see on page 3, lines 158-161 and on page 10, lines 503-511).
Reviewer 3 Report
Comments and Suggestions for Authors
The authors approach an interesting field: The role of endothelium in venous and arterial thrombosis. This issue is worth to be elaborated!
In my opinion there is room for improvement:
i: Looking to the references while studying the manuscript, I wonder if these are really the most recent papers on this topic ? Many (>50!) published more than 10 years ago!
ii: For the reader it is impossible without going in the reference to decide: is this an hyothetical proposal/ possible interpretation or are there study results confirming the statements. The grade of evidence needs to be reflected in the text.
iii: When a thrombus forms in veins or arteries endothelial dysfunction will have to be involved. But endothelial dysfunction may be a primary factor becoming of major importance for thrombus formation, or it may result secondary to thrombus formation. These different aspects should be more clearly discernible.
iv. In contrast to blood cells and plasma components endothelium shows a variable differentiation depending on vascular localisation (not only venous vs arterial but cerebral versus coronary arteries, etc) a point that needs to be mentioned / discussed).
v. Clinical relevance:
Do markers of endothelial dysfunction allow pospectively to evaluate the risk of venous or/and arterial thrombosis? Which markers?
The interaction of inflammation and endothelial function is of particular interest. But there are several links between inflammation and thrombosis beyond endothelial function. More concrete: is there a role of endothelial dysfunction without coincident inflammation. What tell us the available studies.
Author Response
The authors approach an interesting field: The role of endothelium in venous and arterial thrombosis. This issue is worth to be elaborated!
In my opinion there is room for improvement:
i: Looking to the references while studying the manuscript, I wonder if these are really the most recent papers on this topic ? Many (>50!) published more than 10 years ago!
We thank the Reviewer for this comment. We revised the references throughout the manuscript, updating several of them (35 references were removed and updated by 25 new ones), although some other landmark studies were left, even if published in the past. Please see the updated references left highlighted in yellow in the revised manuscript.
ii: For the reader it is impossible without going in the reference to decide: is this an hypothetical proposal/ possible interpretation or are there study results confirming the statements. The grade of evidence needs to be reflected in the text.
We thank the Reviewer for this comment that enables us to improve the manuscript by specifying some important characteristics of the cited studies and/or by better wording information on the level of evidence supporting the main findings throughout the manuscript (Please see the revised manuscript on page 2, lines 87,89,94,101-107; page 3, lines 127-128; page 5, lines 249-253; page 6, lines 28, 293-295; page 7, lines 340-348; page 8, line 390; page 488-494).
iii: When a thrombus forms in veins or arteries endothelial dysfunction will have to be involved. But endothelial dysfunction may be a primary factor becoming of major importance for thrombus formation, or it may result secondary to thrombus formation. These different aspects should be more clearly discernible.
We thank the Reviewer for this insightful comment. We agree with the Reviewer that the issue of endothelial dysfunction (ED) being both the cause and the consequence of thrombosis is an important point that deserves to be discussed more in detail. Accordingly, we have highlighted this aspect in sections 4.2 and 4.3 suggesting also that the timing chosen for ED evaluation can allow the assessment of primary ED, avoiding the confounding effects due to the acute phase of the thrombotic event. Please see the revised manuscript on page 8 (lines 404-408 and lines 419-424).
iv: In contrast to blood cells and plasma components endothelium shows a variable differentiation depending on vascular localisation (not only venous vs arterial but cerebral versus coronary arteries, etc) a point that needs to be mentioned / discussed).
We thank the Reviewer for this insightful comment. Indeed, endothelial cells characteristics vary across different sites of the vascular system and, in the same organ, even across vessels (post-capillaries, venules, arterioles, arteries, small vessels, and large vessels). Moreover, even thrombotic events show peculiar clinical features when occurring at unusual sites (eg. splanchnic vein thrombosis or cerebral vein thrombosis as compared to typical DVT of the lower limbs). Even if we focused on the typical manifestations of thrombosis, we have now mentioned these points in the revised manuscript (please see on page 3, lines 158-161 and on page 10, lines 503-511).
v: Clinical relevance:
Do markers of endothelial dysfunction allow pospectively to evaluate the risk of venous or/and arterial thrombosis? Which markers?
We thank the Reviewer for raising this point. Several biomarkers of ED have been associated with the risk of subsequent thrombosis in clinical studies, such as plasma levels of endocan, endoglin, ADMA and EPCs, and adhesion molecules like ICAM-1, VCAM-1, and E-selectin. These results have been expanded in the revised manuscript on pages 6 lines 319-325, page 7 lines 340-348, lines 358-360 and lines 388-391. However, there has been insufficient evidence so far on the diagnostic accuracy and predictive power of these markers to allow their introduction in common clinical practice.
The interaction of inflammation and endothelial function is of particular interest. But there are several links between inflammation and thrombosis beyond endothelial function. More concrete: is there a role of endothelial dysfunction without coincident inflammation. What tell us the available studies.
We thank the Reviewer for bringing up an important question. Endothelial cells are surely key players in immune thrombosis, where they not only regulate vascular permeability and coagulation but also mediate the inflammatory response, facilitating the interaction between immune cells and the thrombotic process during infections or inflammatory conditions. Nevertheless, endothelial dysfunction can indeed occur independently of inflammation, and this is particularly relevant in the context of thrombosis. Factors such as mechanical stress, oxidative damage, and hypercoagulable states can directly impair endothelial function, leading to thrombosis without the involvement of an inflammatory process [doi: 10.1161/CIRCRESAHA.115.306301; 10.1152/physrev.00029.2006; 10.3390/jcm13020362]. These findings support the idea that endothelial dysfunction and thrombosis are not always linked to systemic inflammation. In addition, studies performed by characterizing in vitro ECFCs derived from patients who experienced unprovoked VTE events showed that a primary ED is present and thus could be involved in the pathogenesis of thrombotic events [ref: 10.1093/cvr/cvz131; 10.4081/btvb.2022.57]. We have added this point in the revised manuscript on page 3, lines 184-189 and discussed an additional citation on page 8, lines 404-408.
Round 2
Reviewer 1 Report
Comments and Suggestions for Authors
I do not have further comments
Author Response
No further comments were raised by Reviewer #1
Reviewer 3 Report
Comments and Suggestions for Authors
The authors provide an improved manuscript.
Several statements need a reference (at least: Lines 178-183; 458-461; 484-486)
There are inconsistencies in the use of the abbreviations - others are not explained (hiPSC) - and the capitalization ("Endoglin / endoglin) within the text.
Author Response
Comment 1. The authors provide an improved manuscript.
We thank the Reviewer for the endorsement of our work.
Comment 2. Several statements need a reference (at least: Lines 178-183; 458-461; 484-486)
We added the references to the indicated sentences. Please see the revised manuscript on page 4, lines 187, 190; page 10, lines 47, 503.
Comment 3. There are inconsistencies in the use of the abbreviations - others are not explained (hiPSC) - and the capitalization ("Endoglin / endoglin) within the text.
We thank the reviewer for identifying these inconsistencies in the use of the abbreviations. We have checked throughout the manuscript and revised accordingly (Please see the revised manuscript on page 2, line 104; page 9, lines 515,516; page 10, lines 519,524,526,527)
We provided explanation for hiPSC (please see the revised manuscript on page 10, lines 493,494) and corrected capitalization of Endoglin (please see the revised manuscript on page 7, line 335; page 8, lines 368, 374; page 10, line 497).